# Investigation of Recent Metaheuristics Based Selective Harmonic Elimination Problem for Different Levels of Multilevel Inverters

Satılmış Ürgün [1] , Halil Yiğit [2,*] and Seyedali Mirjalili [3,4]

1   Faculty of Aeronautics and Astronautics, Kocaeli University, 41380 Kocaeli, Turkey
2   Department of Information Systems Engineering, Kocaeli University, 41001 Kocaeli, Turkey
3   Centre for Artificial Intelligence Research and Optimization, Torrens University Australia (Brisbane), Fortitude Valley, QLD 4006, Australia
4   Yonsei Frontier Lab, Yonsei University, Seoul 03722, Republic of Korea
*   Correspondence: halilyigit@kocaeli.edu.tr

**Abstract:** Multilevel inverters (MLI) are popular in high-power applications. MLIs are generally configured to have switches reduced by switching techniques that eliminate low-order harmonics. The selective harmonic elimination (SHE) method, which significantly reduces the number of switching, determines the optimal switching moments to obtain the desired output voltage and eliminates the desired harmonic components. To solve the SHE problem, classical methods are primarily employed. The disadvantages of such methods are the high probability of trapping in locally optimal solutions and their dependence on initial controlling parameters. One solution to overcome this problem is the use of metaheuristic algorithms. In this study, firstly, 22 metaheuristic algorithms with different sources of inspiration were used to solve the SHE problem at different levels of MLIs, and their performances were extensively analyzed. To reveal the method that offers the best solution, these algorithms were first applied to an 11-level MLI circuit, and six methods were determined as a result of the performance analysis. As a result of the evaluation, the outstanding methods were SPBO, BMO, GA, GWO, MFO, and SPSA. As a result of the application of superior methods to 7-, 11-, 15-, and 19-level MLIs according to the IEEE 519—2014 standard, it has been shown that BMO outperforms in 7-level MLI, GA in 11-level MLI, and SPBO in 15- and 19-level MLIs in terms of THD, while in terms of output voltage quality, GA in 7-level MLI, BMO in 11-level MLI, GA and SPSA in 15-level MLI, and SPSA in 19-level MLI come forward.

**Keywords:** metaheuristic algorithms; multilevel inverter (MLI); selective harmonic elimination (SHE); total harmonic distortion (THD)

## 1. Introduction

Multilevel inverters (MLIs) are electronic circuits used to obtain high-quality and efficient output voltages by using semiconductor power switches without the need for passive elements. MLIs are widely used in applications such as medium- and high-power UPS, high-voltage DC transmission, and variable frequency drives. The notable disadvantages of MLIs are their reliance on isolated power supplies as well as the complexity of converter design and switching control circuits [1,2].

PWM signals can be generated for both fundamental frequency switching and high-frequency switching. The sinusoidal PWM, carrier-based PWM, and space-vector PWM methods provide quality output voltage thanks to the high-frequency switching signals they produce while reducing harmonic components [3]. However, high-frequency switching causes the switching losses to increase and the inverter efficiency to decrease. On the other hand, the selective harmonic elimination pulse width modulation (SHEPWM) technique produces switching signals at the fundamental frequency of switching. In this

method, decreasing the switching frequency causes the switching losses to decrease and the efficiency to increase [4,5]. In SHEPWM, the Fourier series representing the fundamental components and harmonics of the MLI output voltage waveform is formulated with a set of nonlinear transcendental equations. In order to obtain switching signals, iterative methods, such as Newton–Raphson (NR) were first used to solve these equations. However, the NR method has two major drawbacks. These are the operations of the method in a certain modulation range and the difficulty in determining the initial values of the switching range [6,7]. If these two conditions are not chosen well, the method will fail to converge and get stuck in the local optimum. Methods such as the polynomial synthesis method, power method, Walsh function, and homotopic algorithm can also be exploited for solving SHE equations. Their dependence on initial conditions and computational complexity are the major drawbacks of the methods [8–11]. To overcome these problems, many meta-heuristic methods, in which natural processes are imitated, have been used in recent years. Optimal targets can be achieved over the entire range of the modulation index using such methods. Metaheuristics minimize the SHE equations to the most optimal possible values. In applications where the output voltage is trying to be kept constant or the harmonics need to be eliminated independently of the output voltage, these equations are solved by metaheuristic methods, which allow reaching the global optimum with higher precision.

In recent years, swarm-based, evolutionary theory-based, physics-based, human-based, and mathematical-based metaheuristic methods have been applied to solve SHEPWM equations. These metaheuristics can be explained as follows:

Differential evolutionary (DE) [12] and genetic algorithms (GAs) [13] are the best-known evolution-based algorithms. Particle swarm optimization (PSO) [14], ant colony optimization (ACO) [15], bat algorithm (BA) [16], artificial bee colony (ABC) [17], cuckoo search algorithm (CSA) [18], grey wolf optimizer (GWO) [19], dragonfly algorithm (DA) [20], firefly algorithm (FA) [21], flower pollination algorithm (FPA) [22], moth-flame optimizer (MFO) [23], ant lion algorithm (ALO) [24], whale optimization algorithm (WOA) [25], Harris hawks optimization (HHO) [26], Salp swarm algorithm (SSA) [27], marine predator algorithm (MPA) [28], black widow optimization algorithm (BWOA) [29], bacterial foraging algorithm (BFA) [30], dingo optimization algorithm (DOA) [31], and artificial hummingbird algorithm (AHA) [32], which are the popular swarm algorithms, teaching learning based optimization (TLBO) [33] and imperialist competition algorithm (ICA) [34], which are popular human-based algorithms inspired by any event or behavior inside the event-based class. Archimedes optimization algorithm (AOA) [35], artificial electric field algorithm (AEFA) [36], atom search optimization (ASO) [37], equilibrium optimizer (EO) [38], flow direction algorithm (FDA) [39], gravitational search algorithm (GSA) [40], and multiverse optimizer (MVO) [41] are physics-based algorithms inspired by a physical phenomenon. The sine cosine algorithm (SCA) [42], which is the most popular of the math-based algorithms, harmony search (HS) [43], crystal structure algorithm (CryStAl) [44], and colonial competitive algorithm (CCA) [45], which are the other metaheuristic algorithms. In all the above-mentioned studies, it has been shown that the applied algorithms can effectively remove unwanted harmonics from the SHEPWM control strategy. A broad literature summary of the studies on this subject is given below.

Total harmonic distortion (THD) minimization and selective harmonic elimination (SHE) control techniques have been developed to determine switching angles that minimize and eliminate low-order harmonics. In the first technique, THD is reduced without focusing on certain harmonics. On the other, the harmonics are eliminated until the degree of freedom. In three-phase MLIs, harmonics, an odd multiple of three, are automatically eliminated. However, THD minimization is more challenging in single-phase inverters. For this reason, a single-phase MLI structure was applied in this study.

In the one-phase topology introduced in [46], a structure that creates a lower blocking voltage on the switches and reduces the number of switching elements is proposed. The single-phase cascaded inverter topology, which determines the switching moments using the combination of the THD and SHE methods, is presented in [47]. Ref. [48] considers a

single-phase, five-level PUC inverter with a 1:1 DC-link ratio developed for solar PV systems. The switching signals of the inverter are produced by an ANN-based controller. The authors propose a selective harmonic elimination-based novel fast convergent homotopy perturbation technique as the switching method in the single-phase cascade MLI topology in [49]. In [50], a case study of three-level neutral-point-clamped inverters is examined to demonstrate the proposed SHEPWM formulation and the coupling effects between the common mode voltage (CMV) reduction and capacitor voltage balancing targets of multilevel power converters. In [51], switching signals were produced for a nine-level DC-link multilevel inverter using nearest-level modulation and Gaussian selective harmonic elimination techniques. In [52], a new pulse amplitude width modulation (PAWM) technique is implemented to obtain switching signals for a three-phase and 11-level cascaded H-bridge multilevel inverter. Ref. [53] presents a new analytical procedure for selective harmonic elimination to determine switching states for the five-level inverter used at the fundamental frequency in distributed generation systems. In [54], a single-phase cascaded multilevel inverter using a new base unit topology, in which the number of switches is reduced, is suggested. An algebraic method based on Groebner bases and symmetric polynomial theories is presented in [55] to ascertain the switching angles with harmonic elimination in classical MLI structures.

In [56], the switching moments for seven- and nine-level MLIs were optimized using APSO-GA, PSO, APSO, bee algorithm (BA), differential evolution (DE), synchronous PSO, and TLBO optimization techniques. This study focuses on the APSO-GA technique and compares it with other methods, but no comparison is performed between the methods in terms of performance. Another study in which the standard colonial competitive algorithm (CCA) technique is used, and the results are compared with GA and PSO techniques is presented in [57]. A study using the flower pollination algorithm (FPA) technique to determine the switching angles for symmetrical and asymmetrical switched-diode multilevel inverters (SMLI) is suggested in [58]. The results acquired with FPA are compared with those obtained with PSO, TBLO, and CSA methods. In [59], the general mathematical solution method is employed as the SHE method for symmetric and asymmetric MLI circuits. Although the advantages of the method are listed, no comparison with other methods has been performed. A study in which the stochastic THD (STHD) strategy is exploited as the SHE method for multilevel flying capacitor inverter (MFCI) structures is presented in [60]. The study presented in [61] proposes a modified particle swarm optimization (MPSO) method for harmonic elimination in three-phase, eleven-level hybrid cascaded multilevel inverter (HC-MLI) structures. The results are compared with PSO and GA methods. A study, in which the modified grey wolf optimization (MGWO) technique is identified as the switching technique in classical 11-level hybrid cascaded MLIs, is proposed in [62]. The performance of the MGWO method evaluates the GA, PSO, and GWO techniques. A new topology is introduced for nine-level single-phase MLI in [63]. The hybrid gravity search algorithm (GSA)-based SHE technique is applied to calculate the switching moments. The performance evaluation of the method is carried out with the CGA, PSO, and GSA methods. A study using the SHEPWM technique in a new five-level single-phase inverter structure is given in [64]. Here, the optimum switching angles are derived by minimizing the THD function. No comparisons have been carried out using different methods. Making more than one switch at the same level makes the study unique. Since it deals with THD as a fitness function, the output voltage is not included in the optimization problem. The study, using an opposition-based quantum bat algorithm (OQBA) as a SHEPWM technique in classical three-phase cascade MLI circuits, is presented in [65]. GA, WOA, the improved immune algorithm (IIA), bacterial foraging (BP), differential harmony search (DHS), and PSO techniques were applied to evaluate the performance of the method.

Another study, in which the switching moments are determined by the TLBO method in a classical one-phase 5-level MLI, is given in [66]. To test the performance of the proposed method, comparisons were executed with the GA, ABC, ICA, HS, DE, ACO, and PSO. In [67], the switching angles that will eliminate harmonics for a 5- and 7-level

one-phase classical cascade MLI are obtained using the generalized pattern search (GPS) algorithm. The validity of the proposed method is tested with GA. The study, in which the heterogeneous comprehensive learning particle swarm optimization (HCLPSO) method is used to find the optimum switching angles in classical 3-phase H-bridge eleven-level multilevel inverters, is given in [68]. The success of the HCLPSO is compared with the gravitational search algorithm (GSA) and differential search algorithm (DSA) methods. A study, in which the artificial jellyfish search (AJS) algorithm-based SHE method is used to eliminate low-order harmonics in classical H-bridge MLI with five, seven, and nine levels, is presented in [69]. The results obtained from the proposed method are collated with the differential evolution (DE) and GA methods. In [70], the modified quantum particle swarm optimization (MQPSO) method is applied to produce the optimal switching angle for SHEPWM signals. The bioinspired black widow optimization algorithm (BWOA), another alternative method used to solve the SHE problem for three-phase, eleven-level cascade H-bridge MLI, is presented in [71]. The success of the method has been evaluated without any comparison.

Ref. [72] presents a comparison of SSA, multiverse optimization (MVO), and PSO methods for THD minimization in single-phase, seven-level cascaded H-bridge MLIs. It is emphasized that the MVO method gives better numerical results. A study using the multiobjective whale optimization algorithm (MOWOA) to determine the optimum switching moments in asymmetrical half-cascaded multilevel inverters is presented in [73]. In [74], the switching moments required for SHE in three-phase, 11-level cascaded H-bridge MLI are optimized using the marine predator algorithm (MPA) method. The performance evaluation is accomplished by the TLBO, FPA, PSO, and PSO-GWO algorithms. In [75], the shuffled frog leaping algorithm (SFLA) method is implemented to reveal the switching moments in the three-phase, 11-level cascade MLI. Comparisons with other methods have not been conducted; only the application of the method has been explained. Ref. [76] employs the GA-based bio-inspired AI algorithm to determine the optimum angle values in classical single-phase cascade H-bridge MLIs.

Metaheuristic algorithms have a level of complexity due to their nature. On the other hand, as the MLI level increases, the complexity of the fitness function and the number of switching elements increase significantly. In this context, many metaheuristic methods have been applied to the SHEPWM method in the literature in recent years. In this study, the relationship between different algorithms and inverter levels has been extensively examined in an effort to find the most suitable metaheuristic algorithm for the SHEPWM method in MLIs. Complexity, inverter level, THD, and reaching the desired output voltage are presented in detail.

When the studies in the literature are examined, it can be concluded that there is no algorithm that can solve every optimization problem efficiently and effectively. While the optimal solution can be obtained using one method for a particular problem, the solution may not be reached for another problem. Therefore, to settle the optimum switching angle for the SHE problem, it is important to investigate and select new generation optimization algorithms that minimize the limitations such as optimal solution speed, convergence rate, and computation time.

In our previous study, the performance of MFO, SCA, DA, AHA, ASO, EO, FDA, and AOA optimization algorithms on the 11-level MLI design was evaluated in terms of convergence rate, iteration time, THD minimization, and obtaining the desired output voltage to determine optimum switching moments [77]. In the current study, the switching moments of MLIs are analyzed by using new methods that have not been implemented in the optimization of the SHE problem before, such as BMO, GBO, GTO, JS, LA, MGA, PO, POA, PPA, SPBO, SPSA, SSO, and WHO. In addition, a comprehensive analysis is performed with widely used metaheuristics such as SSA, WOA, GWO, PSO, MFO, SCA, DOA, TLBO, and GA for MLIs of different levels. The contribution of this study to the literature is presented below:

- The switching moments obtained by the SHE method are applied to an MLI structure that exploits fewer elements than the classical cascade MLI structures, in which a large number of switching elements are used because the cost will increase rapidly with the level of a classical MLI.
- In order to solve the non-linear transcendental equation, set, new generation metaheuristic algorithms that have not been applied before in the literature are examined.
- By examining MLIs with different levels, the effect of increasing the number of levels on the cost, THD minimization, the quality of the desired output voltage, the computing time (related to computational complexity), and the convergence rate of metaheuristic algorithms are revealed in detail.
- Each method cannot produce an optimal solution over the entire modulation index range. Therefore, comparing the performances of algorithms within the entire modulation index range, such as $0.1 \leq M_i \leq 1.1$, and a certain modulation index range, such as $0.4 \leq M_i \leq 0.9$, will allow the determination of the most efficient method.
- In this study, the most comprehensive evaluation of classical and current metaheuristic methods, which are run with initial parameters such as the maximum number of iterations, search range limits, and the number of search agents, is applied to solve the SHE problem. The results obtained contribute to the field of metaheuristics in terms of better analysis of the performances of the algorithms.

The remainder of the presented work is organized as follows: The following section introduces the concept of multilevel inverters and describes the SHEPWM technique. Section 3 first states an explanation of the metaheuristics used in this study. Then, the simulation results and analysis of suggested metaheuristic algorithms are presented. The discussion is presented in Section 4. In the last section, conclusions and future studies are given.

## 2. Multilevel Inverter

### 2.1. Output Voltage Waveform of Multilevel Inverter

In the classical cascaded H-bridge MLI topology (CHB-MLI), two or more H-bridge cells are connected in series. Since the number of switching elements is high in classical CHB-MLI structures, many studies are trying to eliminate this disadvantage with alternative topologies. As a result, the purpose of all circuits is to create a desired output voltage by adding DC voltage sources to each other, as seen in Figure 1b. The number of DC sources used represents the level of MLIs expressed in (1).

$$\text{Level of MLI} = 2S + 1. \tag{1}$$

where $S$ provides the number of DC sources or the number of switches per quarter.

In this study, an inverter given in Figure 1a is proposed to reduce the number of switching elements in MLI topologies. The switches forming each level and a series diode are connected between the source and the H-bridge. A switching element is connected in reverse parallel between the cell and the source of freewheeling currents. The free toggle switch is switched with the cutoff of the elements in the H-bridge and is taken to the cutoff after a certain period.

There are $S$ variables up to $\alpha_1$, $\alpha_2$, ... , and $\alpha_S$ in MLI at a certain level and the stepping output voltage is obtained using the Fourier series expansion, $f(t)$, given in (2).

$$f(t) = a_0 f(t) = a_0 + \sum_{n=1}^{\infty}(a_n cos(n\omega t) + b_n sin(n\omega t)) \tag{2}$$

where $n$ is the harmonic number, and $a_n$ and $b_n$ are the even and odd component amplitudes of the $n$th harmonic, respectively. $a_0$ is the DC coefficient. In this equation, considering

the quarter-wave symmetry, the coefficients $a_0$, and even harmonics will be zero and the equation will turn into Equation (3).

$$b_n = \begin{cases} \frac{4V_{dc}}{n\pi} \sum_{k=1}^{N}(-1)^{k+1}\cos(n\alpha_k), & for\ odd\ n \\ 0, & for\ even\ n \end{cases} \tag{3}$$

If Equation (3) is rearranged, first, Equation (4) is obtained and finally, the output voltage ($V_0$) is formed as in Equation (5).

$$b_n = \begin{cases} \frac{4V_{dc}}{n\pi} \sum_{k=1}^{N}\cos(n\alpha_k), & for\ odd\ n \\ 0, & for\ even\ n \end{cases}, \tag{4}$$

$$V_O = \sum_{n=1}^{\infty}\left[\frac{4V_{dc}}{n\pi} \sum_{k=1,3,5...}^{N}\cos(n\alpha_k)\right]\sin(n\omega t). \tag{5}$$

where $V_{dc}$ is the nominal DC voltage, $n$ is the number of switching angles, and $\alpha_k$ is the firing angles calculated in the order ($0 < \alpha_1 < \ldots < \alpha_S \leq \pi/2$).

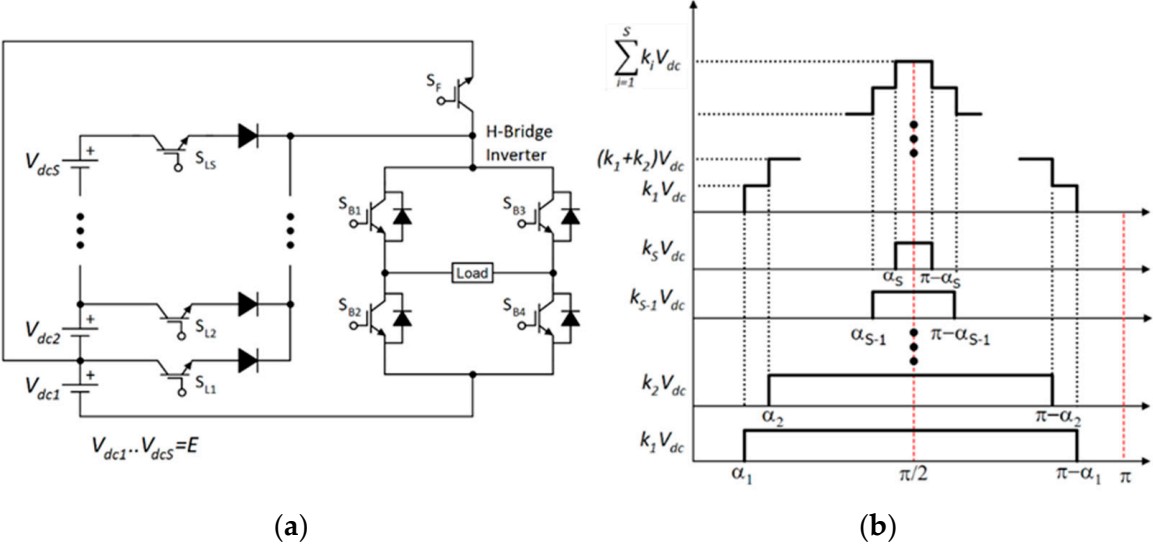

(**a**)　　　　　　　　　　　　　　　　(**b**)

**Figure 1.** (**a**) MLI structure; (**b**) ladder waveform of seven-level inverter.

## 2.2. Selective Harmonic Elimination PWM Technique

The SHEPWM method has two main tasks: eliminating the selected harmonics and adjusting the fundamental component of the output voltage to the desired level. The DC voltage level forms the steps of the output voltage. The number of DC voltage levels is the same as the number of switching moments in a quarter. If $S$ switching angles are exploited, $S$ degrees of freedom are obtained. One of these degrees of freedom tries to hold the output voltage's fundamental component constant, while the others eliminate the selected harmonics. Assuming the DC sources are equal, the equation sets given in (6) are acquired in an MLI with $2S+1$ levels.

$$\left.\begin{aligned} M_i &= \tfrac{1}{S}(\cos(\alpha_1) + \cos(\alpha_2) + \ldots + \cos(\alpha_k)) \\ 0 &= \cos(3\alpha_1) + \cos(3\alpha_2) + \ldots + \cos(3\alpha_k) \\ &\qquad\qquad\vdots \\ 0 &= \cos((2k-1)\alpha_1) + \cos((2k-1)\alpha_2) + \ldots + \cos((2k-1)\alpha_k) \end{aligned}\right\}, \tag{6}$$

where $\alpha$ is the switching moments in the quarter period and is defined by (6). The modulation index ($M_i$) is defined by (7).

$$0 \leq \alpha_1 \leq \alpha_2 \leq \ldots \leq \alpha_k < \frac{\pi}{2} \tag{7}$$

$$M_i = \frac{\pi V_{des}}{4 S V_{dc}}, 0 < M_i \leq 1.1. \tag{8}$$

where $V_{des}$ is the desired or calculated value of the output voltage and $S$ is the switching number. For example, in a 7-level MLI, two unwanted harmonics can be eliminated while keeping the output voltage constant. Thus, Equation (6) becomes the set of equations in (9).

$$\left.\begin{array}{l} M_i = \frac{1}{3}\left(\cos(\alpha_1) + \cos(\alpha_2) + \cos(\alpha_3)\right) \\ 0 = \cos(3\alpha_1) + \cos(3\alpha_2) + \cos(3\alpha_3) \\ 0 = \cos(5\alpha_1) + \cos(5\alpha_2) + \cos(5\alpha_3) \end{array}\right\}. \tag{9}$$

Equation (9) can be similarly obtained for 11-, 15-, and 19-level MLIs. The fitness function is achieved by considering Equations (3) and (6). In this study, since the output voltage and THD value of MLI are optimized together, the fitness function in Equation (10), which is widely used in the literature [55,58,63], which combines different constraints, is considered.

$$Fitness\ function\ (f) = \min\left[\left|A\ \frac{V_{des} - V_1}{V_{des}}\right|^4 + \Sigma_{s=2}^{S} \frac{1}{h_s}\left|B\ \frac{V_{hs}}{V_1}\right|^2\right]. \tag{10}$$

where $V_{des}$ is the desired fundamental voltage and $V_1$ represents the fundamental output voltage of MLI. $h_s$ and $V_{hs}$ are the order and amplitude of the $s$th harmonic, respectively, e.g., $h_2 = 3$, $h_3 = 5$, and $h_5 = 11$. The variables ($\alpha$) are limited by Equation (7).

For the 7-level MLI structure, the solution of Equation (9) provides three switching angles in the range of 0–$\pi/2$. In MLI, the number of switching moments will increase as the number of levels increases. Hence, the number of nonlinear transcendental equations increases. In this case, it is more difficult, more costly, and time-consuming to converge to the optimal result. One of three solutions was used to hold the fundamental component of the output voltage constant. The other two solutions were exploited to minimize low-value harmonic components. When interpreted in terms of the fitness function, the fitness function has the output voltage and THD components. In this study, both components were optimized simultaneously. A trade-off is procured by varying the coefficients of the output voltage and THD components. There are some important difficulties specific to this problem. These are the determination of the fitness function's coefficients and initial values of the angles. If the upper and lower angle limits in the search space are narrow, the algorithms will not converge to the optimum result, and the output voltage will not remain constant at the desired value.

## 3. Implementation of the Harmonic Problem

A summary of the metaheuristic algorithms used in the study to acquire the minimum of the fitness function given by Equation (10) is given in Table 1. GA, SSA, WOA, GWO, PSO, MFO, SCA, DOA, and TLBO have been applied to the optimization of the SHE problem in the literature, while BMO, GBO, GTO, JS, LA, MGA, PO, POA, PPA, SPBO, SPSA, SSO, and WHO algorithms have not been implemented before. Since it will significantly increase the number of pages, instead of explaining the applied algorithms in detail, brief information about the inspired natural phenomenon is given.

**Table 1.** A summary of the metaheuristic algorithms.

| Optimization Method | Inspirer |
|---|---|
| Particle Swarm Optimization (PSO) | Inspired by a school of fish or a flock of birds moving in a group [14] |
| Genetic Algorithm (GA) | Inspired by the natural evolutionary process [12] |
| Whale Optimization Algorithm (WOA) | Inspired by the air bubble behavior of humpback whales use while hunting [25] |
| Grey Wolf Optimization (GWO) | Inspired by the hunting behavior and social leadership of grey wolves [19] |
| Moth Flame Optimization (MFO) | Inspired by the transverse orientation behavior of moths [23] |
| Sine Cosine Algorithm (SCA) | Inspired by the concept of trigonometric sine and cosine functions [42] |
| Teaching Learning based Optimization (TLBO) | Inspired by the teaching and learning behavior in a classroom [33] |
| Sparrow Search Algorithm (SPSA) | Inspired by sparrows' group erudition, foraging, and anti-predation behaviors [78] |
| Student Psychology based Optimization (SPBO) | Inspired by the psychology of students striving to be the best student in the class [79] |
| Barnacles Mating Optimizer (BMO) | Inspired by the mating process of barnacles [80] |
| Dingo Optimization Algorithm (DOA) | Inspired by the social, cooperative, and hunting action of dingoes [31] |
| Gradient-Based Optimizer (GBO) | Inspired by Newton's method that integrates both the gradient search rule and local escaping operator [81] |
| Gorilla Troops Optimizer (GTO) | Inspired by gorilla troops' social intelligence in nature [82] |
| Jellyfish Search Optimizer (JS) | Inspired by the act and foraging behavior of jellyfish in the ocean [83] |
| Lichtenberg Algorithm (LA) | Inspired by the Lichtenberg figure patterns [84] |
| Material Generation Algorithm (MGA) | Inspired by the configuration of chemical compounds and reactions in the production of new materials [85] |
| Political Optimizer (PO) | Inspired by the mathematical mapping of the multistage process of politics [86] |
| Peafowl Optimization Algorithm (POA) | Inspired by the group foraging behavior of the peafowl swarm [87] |
| Parasitism—Predation algorithm (PPA) | Inspired by the multi-interactions between cuckoos, crows, and cats [88] |
| Smell Agent Optimization (SAO) | Inspired by the relationship between a smell agent and an object that vaporizes a small molecule [89] |
| Sperm Swarm Optimization (SSO) | Inspired by sperm-ovum interactions in the fertilization procedure [90] |
| Wild Horse Optimizer (WHO) | Inspired by the social life behavior of wild horses [91] |

When the literature was reviewed, no study was found to determine the $A$ and $B$ coefficients in the fitness function given in (10). In many studies, the $A$ and $B$ coefficients are taken as 100 and 50, respectively [56–58,61,63,66–69,74]. In some studies, only the right part of Equation (10) is included in the calculation as a fitness function [59,64,65]. In addition, the same coefficients $A$ and $B$ are used. As the values of the $A$ and $B$ coefficients change,

the weights of the first and second terms of Equation (10) also vary. If the coefficient $A$ is selected to be larger, the stability of the output voltage increases. In the opposite case, the elimination of harmonics gains weight, and the THD value decreases further. In the presented study, the coefficients $A$ and $B$ were set to 50 and 5, respectively. The aim is to optimize the output voltage, which has a high weight in the fitness function.

The simulation of the multilevel inverter given in Figure 1a using selected metaheuristic algorithms is performed in a MATLAB environment. Simulink, FFT, and curve-fitting tools are employed. The switching angles obtained after compiling the optimization algorithms are applied to the different levels of the MLI to produce an output voltage with the desired number of voltage levels. The specifications of the parameters used in the simulations are given in Table 2.

**Table 2.** Simulation parameters.

| S.No | Parameters | Specification |
|:---:|:---:|:---:|
| 1 | Number of Levels | 7, 11, 15, and 19 |
| 2 | Voltage Source (DC) | 100, 60, 42.85, and 33.33 V |
| 3 | Fundamental frequency | 50 Hz |
| 4 | Load | R = 50 Ω, L = 20 mH |

The voltage sources given in Table 2 for each level are selected to obtain the same voltage value at the output of each level. An inductive load is connected to the inverter output. The same harmonic values will also be seen in different selected frequencies and sources. The THD value and the fundamental components of the output voltage are obtained using the FFT tool. Subsequently, the obtained values are converted into graphics using the curve-fitting tool.

This study consists of two stages. The first stage includes examining the performances of 22 different optimization methods in terms of convergence rate, a single iteration time, and robustness. The THD minimization performances of the methods were examined according to the IEEE 519—2014 standard. Finally, the performance of the methods in producing the desired output voltage is evaluated by considering the 11-level MLI fitness function. In the second stage, performance analysis is carried out for different levels with six metaheuristic algorithms that stand out according to the performance and THD minimization criteria. Once and for all, it was observed that the performance of the algorithms changed according to the inverter level, and a different optimization algorithm was proposed for each level. The two stages of the process are briefly described in Figure 2.

Equations (11) and (12) depict the error value ($e_k$) in any modulation index, and the total THD error value over the entire modulation index range, respectively.

$$e_k = Ithd_{Mk} - Ithd_{Mk(\min)} \tag{11}$$

$$TIthd_e = \sum_{M_{k=0.1}}^{1.1} e_k \tag{12}$$

where $Ithd_{Mk(\min)}$ is the THD minimum value obtained in all algorithms for each $M_i$. $Ithd_{Mk}$ is the THD value of each algorithm.

The output voltage is defined in terms of per unit (*pu*) as given in (13).

$$V_{pu} = \frac{V_1}{V_{des}} \tag{13}$$

where $V_1$ and $V_{des}$ are the values of output voltage and desired output voltage, respectively.

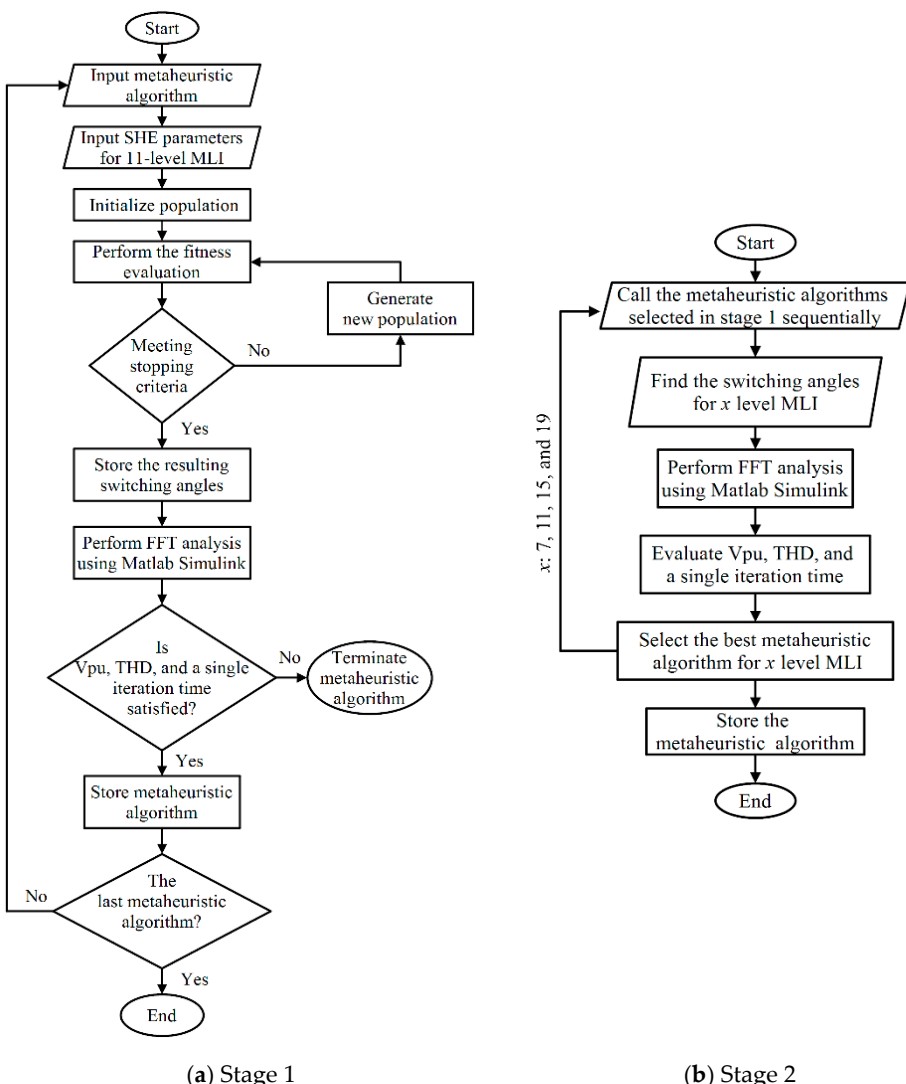

(**a**) Stage 1                                    (**b**) Stage 2

**Figure 2.** Flowchart of the process (**a**) stage 1; (**b**) stage 2.

To find the *pu* deviation in a modulation index ($M_i$), $V_{pu}$ is subtracted from 1. Then, the sum of *pu* errors in all $M_i$ is obtained using Equation (14).

$$TVpu_e = \sum_{M_k=0.1}^{1.1} |1 - Vpu_k| \tag{14}$$

where $TVpu_e$ is the total $V_{pu}$ error.

Considering that it is more realistic to operate the modulation index of MLIs in the range of $0.4 < M_i < 0.9$, the performance analysis was also carried out within this range because, in case $M_i > 0.9$, the output voltage approaches the square waveform. The control parameters of the algorithms are tabulated in Table 3. The parameters in the table are set to the values used in the literature and recommended in the main article of each algorithm. The size of the search agents, i.e., the population in the search space, is tuned to 100. The maximum number of iterations of 500 is chosen to plot the convergence curves of the methods. The best cost value for each algorithm is obtained by running all methods independently 500 times. The algorithm that provides the most statistically optimal cost is selected.

**Table 3.** The parameter settings of the optimization algorithms.

| Method | Parameter | Value |
| --- | --- | --- |
| GWO | Convergence parameter $a$ <br> $r_1, r_2$ | linearly decreased from 2 to 0 <br> [0, 1] |
| SSA | $c_2, c_3$ | [0, 1] |
| WOA | Convergence parameter $a$ | decreases linearly from 2 to 0 |
| SCA | $r_1$ <br> $r_2, r_3, r_4$ | decreases linearly from 2 to 0 <br> $[0, 2\pi]$, [0, 2], [0, 1] |
| PSO | cognitive coefficient <br> social coefficient <br> inertia constant | 2 <br> 2 <br> decrease from 0.9 to 0.2 |
| MFO | convergence constant <br> spiral constant $b$ | $[-1, -2]$ <br> 1 |
| GA | type, selection <br> crossover <br> mutation | real coded, roulette wheel <br> probability = 0.8 <br> Gaussian (probability = 0.05) |
| TLBO | Teaching factor $T$ | [1, 2] |
| DOA | $MOP$, sensitivity parameter $\alpha$ <br> control parameter $\mu$ | [0.2, 1], 9 <br> 0.1 |
| BMO | penis length of the barnacle $pl$ | 7 |
| GBO | probability parameter $pr$ | 0.5 |
| GTO | $p, \beta, w$ | 0.03, 3, 0.8 |
| JS | distribution coefficient $\beta$ <br> motion coefficient $\gamma$ | 3 <br> 0.1 |
| LA | addition of a refinement $ref$ <br> stick probability $S$ <br> creation radius $R_c$ | 0.4 <br> 1 <br> 150 |
| MGA | the probabilistic component $e^-$ | Gauss Distribution |
| PO | party switching rate $\lambda$ <br> number of parties, constituencies, <br> candidates in each party $n$ | linearly decreased from 1 to 0 <br> 8 |
| POA | number of peacocks <br> rotation radius $R_s$ <br> $\theta_0, \theta_1$ <br> coefficient $\gamma$ | 5 <br> 0.5 <br> 0.1, 1 <br> 1.5 |
| PPA | the intrinsic growth rate of crows, $r_1$ <br> the death rate of cuckoos, $r_2$ <br> the death rate of cats, $r_3$ <br> $\alpha_1, \alpha_2$ <br> $\beta_1, \beta_2$ <br> $c_1, c_2$ <br> $d_1, d_2$ | 1 <br> 0.1 <br> 0.1 <br> 0.2, 0.25 <br> 0.1 <br> 0.1 <br> 0.01 |
| SPBO | Not Available | |
| SPSA | threshold value $ST$ <br> number of the producers $PD$ <br> number of the danger-perceivers $SD$ | 0.8 <br> 20% <br> 10% |
| SSO | pH value <br> a factor of velocity damping $D$ <br> temperature $T$ | [7, 14] <br> [0, 1] <br> [35.1, 38.5] |
| WHO | crossover percentage, $PC$ <br> stallions' percentage, $PS$ | 0.13 <br> 0.2 |

In Table 4, the most optimal values for an iteration time, output voltage quality, and THD minimization parameters among 22 algorithms are shown in bold. When looking at the entire $M_i$ range in terms of THD, SPBO is the best method, while MGA shows the worst performance. The BMO, GWO, MFO, GTO, TLBO, and GBO methods indicate better performance in terms of THD than others. Although the results are close in terms of $V_{pu}$, MGA, SPBO, WHO, SCA, TLBO, and JS are the prominent methods.

In the range of $0.4 < M_i < 0.9$, SPBO, GA, GWO, MFO, SPSA, and BMO are spectacular methods in terms of THD minimization. JS, MGA, SPBO, WHO, SCA, and TLBO methods outperform in terms of output voltage quality.

When the methods are compared in terms of a single iteration time, it is revealed that JS, MGA, PPA, SPSA, SSA, GWO, MFO, and SCA provide faster results than others. The iteration time of a method is directly related to the computational complexity of the method.

The focus of the presented work is output voltage quality and THD minimization. Most of the methods produce similar results in terms of output voltage. For this reason, the SPBO, BMO, GA, GWO, MFO, and SPSA methods, which provide the most optimal results in terms of THD, are employed in the examination of MLIs of different levels, considering the IEEE 519—2014 standard.

The convergence plots of the selected algorithms at the different levels of MLI are presented in Figure 3. It can be clearly seen that as the level value increases in MLIs, the fitness function convergence values of the metaheuristic methods decrease. This is because more harmonic components are tried to be eliminated with increasing levels, as shown in (6). Therefore, the best cost value will decrease. The best cost value for the 7 levels is obtained by the MFO in the 232nd iteration, followed by SPSA and GA. The worst convergence performance occurs in the GWO method. In 11-level MLI, the SPSA method reaches the best fitness value in the 330th iteration among other methods, while MFO and GA achieve the best cost value in the 210th and 107th iterations, respectively. In 15-level MLI, the best cost value reaches a maximum of $10 \times 10^{-5}$. While all methods arrive at the same best cost value at the end of the 500th iteration, SPSA and MFO converge to the value in the 130th iteration. In the case of 19 levels, all methods converge to the maximum value of $10 \times 10^{-4}$ at the 500th iteration. SPSA, GA, MFO, and SPBO offer the best cost faster than other methods. The method with the worst convergence performance at all levels emerges as GWO.

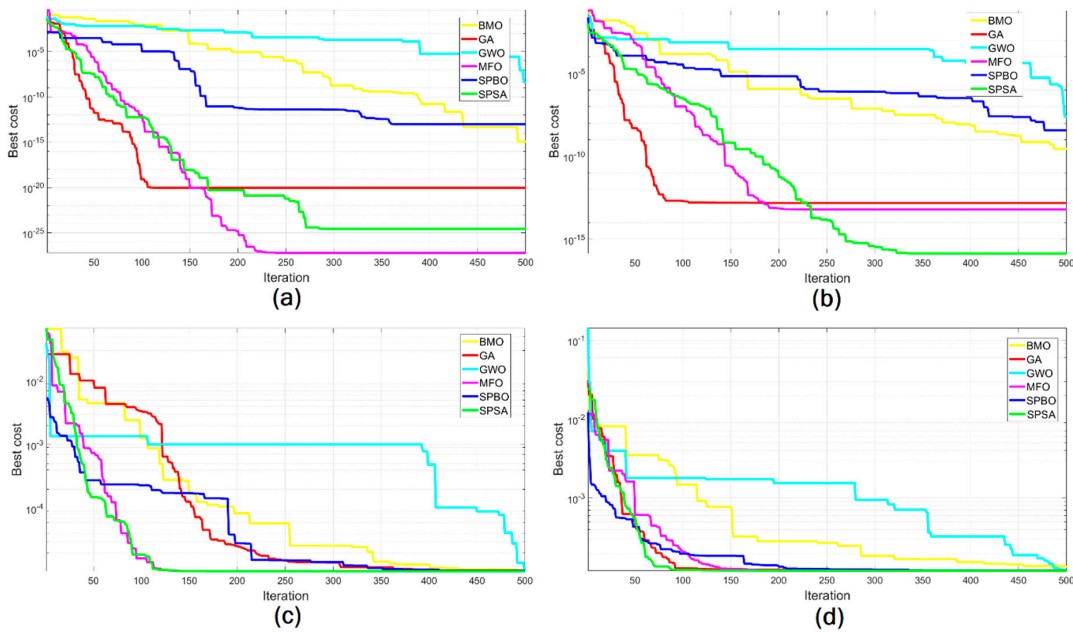

**Figure 3.** Convergence curves of the optimization algorithms for MLI with (**a**) 7-level; (**b**) 11-level; (**c**) 15-level; and (**d**) 19-level.

**Table 4.** Comparison of statistical values of the fitness function's cost, current total harmonic distortions, and output voltages in algorithms applied to the 11-level SHEPWM problem (the prominent values are highlighted in bold).

| Algorithms | A Single Iteration Time (s) | Fitness Function Cost | | | | Total Current THD Error $0.1 < M_i < 1.1$ | Total Output Voltage PU Error $0.1 < M_i < 1.1$ | Total Current THD Error $0.4 < M_i < 0.9$ | Total Output Voltage PU Error $0.4 < M_i < 0.9$ |
| | | Best | Mean | Worst | Std | | | | |
|---|---|---|---|---|---|---|---|---|---|
| BMO | $\mathbf{1.3202 \times 10^{-3}}$ | $\mathbf{2.8741 \times 10^{-10}}$ | $7.3633 \times 10^{-7}$ | $3.0686 \times 10^{-5}$ | $3.4150 \times 10^{-6}$ | **68.96** | 2.08 | **6.12** | 0.55 |
| GBO | $1.6676 \times 10^{-3}$ | $\mathbf{7.7066 \times 10^{-20}}$ | $8.3334 \times 10^{-11}$ | $5.4120 \times 10^{-10}$ | $\mathbf{1.4612 \times 10^{-10}}$ | **70.76** | 2.06 | 8.09 | 0.56 |
| GTO | $1.0636 \times 10^{-3}$ | $\mathbf{9.2000 \times 10^{-23}}$ | $1.7044 \times 10^{-10}$ | $5.4099 \times 10^{-10}$ | $\mathbf{1.9544 \times 10^{-10}}$ | **70.20** | 2.10 | 7.30 | 0.55 |
| JS | $\mathbf{6.3162 \times 10^{-4}}$ | $8.4924 \times 10^{-11}$ | $1.2737 \times 10^{-5}$ | $2.5813 \times 10^{-4}$ | $3.2158 \times 10^{-5}$ | 71.69 | **2.05** | 8.70 | **0.52** |
| LA | $2.9378 \times 10^{-3}$ | $1.0620 \times 10^{-7}$ | $7.9679 \times 10^{-6}$ | $7.7236 \times 10^{-5}$ | $1.3306 \times 10^{-5}$ | 82.00 | 2.13 | 12.86 | 0.55 |
| MGA | $\mathbf{6.0866 \times 10^{-5}}$ | $2.7317 \times 10^{-2}$ | $1.8867 \times 10^{-1}$ | $3.9264 \times 10^{-1}$ | $8.0063 \times 10^{-2}$ | 131.04 | **1.76** | 18.89 | **0.51** |
| PO | $1.4118 \times 10^{-3}$ | $1.6449 \times 10^{-17}$ | $6.9237 \times 10^{-11}$ | $5.4082 \times 10^{-10}$ | $1.1998 \times 10^{-10}$ | 73.94 | 2.12 | 8.54 | 0.57 |
| POA | $2.8153 \times 10^{-3}$ | $2.9787 \times 10^{-17}$ | $2.1614 \times 10^{-10}$ | $5.4112 \times 10^{-10}$ | $2.1120 \times 10^{-10}$ | 71.57 | **2.05** | 8.59 | 0.53 |
| PPA | $\mathbf{8.8035 \times 10^{-4}}$ | $\mathbf{9.6400 \times 10^{-22}}$ | $2.2238 \times 10^{-10}$ | $7.9866 \times 10^{-10}$ | $\mathbf{2.0503 \times 10^{-10}}$ | **70.92** | 2.06 | 8.03 | 0.54 |
| SPBO | $1.3096 \times 10^{-3}$ | $3.7682 \times 10^{-9}$ | $1.6368 \times 10^{-6}$ | $3.2202 \times 10^{-5}$ | $3.9299 \times 10^{-6}$ | **8.03** | **1.79** | **4.82** | **0.52** |
| SPSA | $\mathbf{8.6769 \times 10^{-4}}$ | $1.3969 \times 10^{-16}$ | $1.3471 \times 10^{-10}$ | $5.3942 \times 10^{-10}$ | $\mathbf{1.8617 \times 10^{-10}}$ | 75.70 | 2.14 | **6.43** | 0.58 |
| SSO | $1.4437 \times 10^{-3}$ | $5.4589 \times 10^{-3}$ | $7.5589 \times 10^{-2}$ | $1.8221 \times 10^{-1}$ | $3.7680 \times 10^{-2}$ | 80.52 | 2.07 | 10.51 | 0.61 |
| WHO | $2.1154 \times 10^{-3}$ | $1.6449 \times 10^{-17}$ | $1.1348 \times 10^{-10}$ | $1.0401 \times 10^{-9}$ | $1.6571 \times 10^{-10}$ | 72.01 | **2.04** | 8.96 | **0.52** |
| SSA | $\mathbf{8.5082 \times 10^{-4}}$ | $1.1341 \times 10^{-13}$ | $1.5287 \times 10^{-10}$ | $5.4140 \times 10^{-10}$ | $1.9452 \times 10^{-10}$ | 70.16 | 2.09 | 7.40 | 0.57 |
| WOA | $1.0745 \times 10^{-3}$ | $9.1607 \times 10^{-7}$ | $1.6122 \times 10^{-4}$ | $2.5524 \times 10^{-3}$ | $3.4030 \times 10^{-4}$ | 71.71 | 2.42 | 7.71 | 0.58 |
| GWO | $\mathbf{6.5974 \times 10^{-4}}$ | $2.6246 \times 10^{-8}$ | $2.1951 \times 10^{-4}$ | $6.5700 \times 10^{-3}$ | $8.4363 \times 10^{-4}$ | **69.09** | 2.13 | **5.67** | 0.60 |
| PSO | $4.6609 \times 10^{-3}$ | $\mathbf{5.3836 \times 10^{-18}}$ | $9.6393 \times 10^{-11}$ | $5.3938 \times 10^{-10}$ | $1.4142 \times 10^{-10}$ | 77.39 | 2.16 | 7.70 | 0.61 |
| MFO | $\mathbf{6.5258 \times 10^{-4}}$ | $6.4324 \times 10^{-14}$ | $8.3350 \times 10^{-11}$ | $5.4035 \times 10^{-10}$ | $\mathbf{1.3430 \times 10^{-10}}$ | **70.00** | 2.14 | **5.69** | 0.56 |
| SCA | $\mathbf{6.2341 \times 10^{-4}}$ | $3.0773 \times 10^{-3}$ | $2.9946 \times 10^{-2}$ | $7.9005 \times 10^{-2}$ | $1.4711 \times 10^{-2}$ | 70.73 | **2.03** | 7.35 | **0.50** |
| TLBO | $1.1286 \times 10^{-2}$ | $9.7908 \times 10^{-15}$ | $3.5904 \times 10^{-7}$ | $1.4051 \times 10^{-5}$ | $1.6061 \times 10^{-6}$ | **70.25** | **2.04** | 7.44 | **0.52** |
| GA | $2.3372 \times 10^{-3}$ | $1.5440 \times 10^{-13}$ | $1.2567 \times 10^{-5}$ | $1.3202 \times 10^{-4}$ | $2.3307 \times 10^{-5}$ | 76.74 | 2.17 | **5.48** | 0.60 |
| DOA | $1.5837 \times 10^{-3}$ | $1.7300 \times 10^{-17}$ | $2.6695 \times 10^{-8}$ | $2.6615 \times 10^{-6}$ | $2.6614 \times 10^{-7}$ | 72.46 | 2.13 | 9.20 | 0.59 |

In Table 5, the performances of selected SPBO, BMO, GA, GWO, MFO, and SPSA methods in terms of output voltage, THD, and a single iteration time in 7-, 11-, 15-, and 19-level MLIs are given. Methods that exhibit the best output voltage quality and THD performance are marked in bold font. To examine the complexity of the fitness function at the MLI level, a single iteration time at the different levels is also calculated.

When the variation of an iteration time with an increase in the level values in Table 5 is investigated, GA emerges as the method with the least variation, with a standard deviation value of $3.8770 \times 10^{-5}$. GA is followed by SPSA ($8.0254 \times 10^{-5}$), MFO ($9.1466 \times 10^{-5}$), GWO ($1.1485 \times 10^{-4}$), BMO ($1.5272 \times 10^{-4}$), and SPBO (0.0011), respectively. The SPBO is the method that undergoes the most changes in terms of a single iteration. It is obvious that the MFO method has the smallest iteration time at all levels in comparison with selected metaheuristics according to output voltage, THD minimization, and a single iteration time, respectively.

**Table 5.** Comparison of the error values of the output voltages ($V_{pu}$), the current total harmonic distortions (*Ithd*), and an iteration time in selected algorithms applied to the 7, 11, 15, and 19 levels of MLI.

| | 7 Level MLI | | | 11 Level MLI | | |
|---|---|---|---|---|---|---|
| | $V_{pu}$ **Error** $0.4 \leq M_i \leq 0.9$ | *Ithd* **Error** $0.4 \leq M_i \leq 0.9$ | **A Single Iteration Time (s)** | $V_{pu}$ **Error** $0.4 \leq M_i \leq 0.9$ | *Ithd* **Error** $0.4 \leq M_i \leq 0.9$ | **A Single Iteration Time (s)** |
| SPBO | 0.0606 | 11.8 | $6.347161 \times 10^{-4}$ | 0.0634 | 7.128 | $1.223483 \times 10^{-3}$ |
| BMO | 0.0580 | 4.96 | $1.125343 \times 10^{-3}$ | 0.0538 | 7.878 | $1.235453 \times 10^{-3}$ |
| GA | 0.0495 | 8.65 | $1.592762 \times 10^{-3}$ | 0.0625 | 3.63 | $1.606038 \times 10^{-3}$ |
| GWO | 0.0623 | 11.59 | $2.764809 \times 10^{-4}$ | 0.0624 | 6.258 | $3.541279 \times 10^{-4}$ |
| MFO | 0.0554 | 11.75 | $2.724207 \times 10^{-4}$ | 0.0571 | 6.618 | $3.213458 \times 10^{-4}$ |
| SPSA | 0.0536 | 15.16 | $6.837816 \times 10^{-4}$ | 0.0737 | 11.158 | $7.411829 \times 10^{-4}$ |
| | **15 Level MLI** | | | **19 Level MLI** | | |
| | $V_{pu}$ **Error** $0.4 \leq M_i \leq 0.9$ | *Ithd* **Error** $0.4 \leq M_i \leq 0.9$ | **A Single Iteration Time (s)** | $V_{pu}$ **Error** $0.4 \leq M_i \leq 0.9$ | *Ithd* **Error** $0.4 \leq M_i \leq 0.9$ | **A Single Iteration Time (s)** |
| SPBO | 0.0661 | 1.54 | $2.120586 \times 10^{-3}$ | 0.0651 | 2.29 | $3.247140 \times 10^{-3}$ |
| BMO | 0.0660 | 4.16 | $1.317429 \times 10^{-3}$ | 0.0673 | 8.36 | $1.487826 \times 10^{-3}$ |
| GA | 0.0643 | 4.28 | $1.651683 \times 10^{-3}$ | 0.0647 | 4.38 | $1.675692 \times 10^{-3}$ |
| GWO | 0.0651 | 4.05 | $4.464614 \times 10^{-4}$ | 0.0654 | 3 | $5.419105 \times 10^{-4}$ |
| MFO | 0.0681 | 3.34 | $3.915422 \times 10^{-4}$ | 0.0664 | 3.45 | $4.830447 \times 10^{-4}$ |
| SPSA | 0.0643 | 3.77 | $8.338084 \times 10^{-4}$ | 0.0642 | 5.34 | $8.553041 \times 10^{-4}$ |

The coefficient $A$ in the fitness function is the coefficient of the output voltage, and this part of the equation is independent of the number of levels. Therefore, as can be seen from Table 5, the output voltage at all levels exhibits approximately the same performance. The $B$ coefficient covers the THD part of the fitness function. As the number of levels changes, new functions are added to this section. Therefore, it is expected that there will be changes in the THD performance. Ideally, as the number of equations of the harmonic components added to the fitness function increases, the THD value decreases. Metaheuristics can provide different responses depending on the increasing search conditions (increasing parameter and population number). In the SPBO method, THD values in the range of $0.4 \leq M_i \leq 0.9$ have been obtained at 11.8, 7.13, 1.54, and 2.29 for the 7, 11, 15, and 19 levels, respectively. The SPBO method reveals the best performance in 15-level MLI. In the BMO method, the THD value is seen as 4.96, 7.88, 4.16, and 8.36 depending on the level increase. The BMO method shows its best performance in the 15-level MLI. In the GA method, the THD value has been procured at 8.65, 3.63, 4.28, and 4.38, depending on the level increase. The GA

method depicts its best performance in 11-level MLI. In the GWO method, the THD value has been calculated as 11.59, 6.26, 4.05, and 3 depending on the level increase. The GWO method exhibits better performance with each added harmonic component to the fitness function and reveals its best performance at the 19th level. On the other hand, the MFO method provides similar results to the GWO method. The best performance is exposed at level 15, with a very small margin compared to the MFO. Finally, although the SPSA method performs worse than the others in terms of THD at levels 11 and 15, it yields a better result at levels 15 and 19.

Figure 4 illustrates the variations of a single iteration time and THD for MLI with 7, 11, 15, and 19 levels, respectively. As can be seen from Figure 4a, a single iteration time in the SPBO method increased approximately five times compared to the one obtained in the 7-level MLI, while the average change was 63%. The average variation in a single iteration time in GA was approximately 2.38%. The variations in other methods have been acquired as 11.82% (BMO), 28.37% (GWO), 24.91% (MFO), and 10.30% (SpSA), respectively. Figure 4b plots the THD convergence curves that change depending on the increasing number of parameters in the search space when using multilevel MLI. Except for BMO, other methods approach a certain THD value with an increase the parameter number. Figure 4b clearly demonstrates the relationship between the increase in the number of parameters (thereby, increasing the number of levels) in the search space and the decrease in THD values.

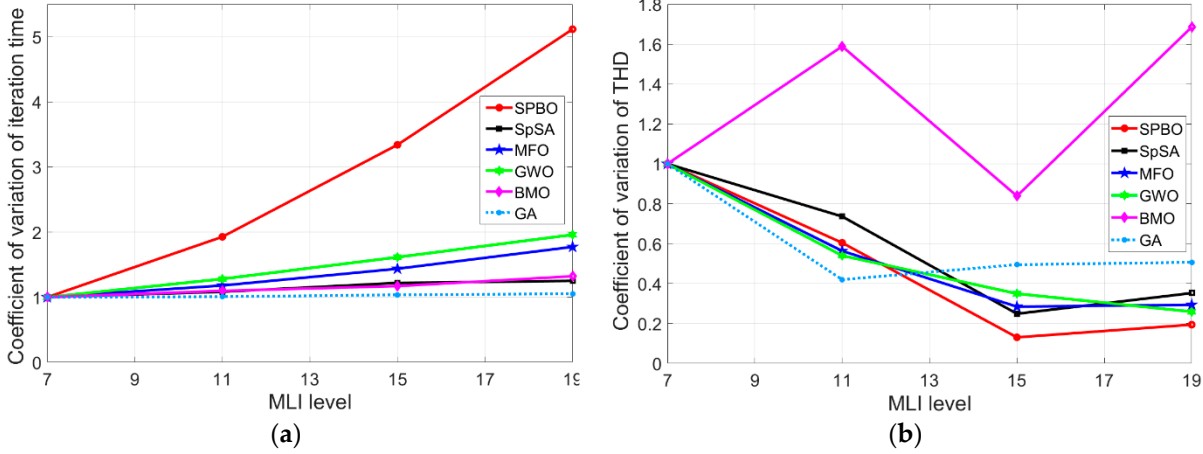

**Figure 4.** Coefficient of variation of (**a**) iteration time; (**b**) THD.

According to the IEEE 519—2014 standard, the THD value should be below 8.0% for devices operating up to 1 kW. When the plots in Figure 5a are evaluated, it is seen that only the BMO and GA methods meet the THD standard in a narrow modulation index range ($0.8 \leq M_i < 0.9$). At this point, it should be taken into account that the output voltage is trying to be kept constant in the 7-level MLI optimization. In the absence of the first term corresponding to the optimization of the output voltage in Equation (10), it is expected that the harmonics will be lower because (10) becomes a THD minimization problem and three available arguments are used. Figure 5b shows the THD performances of the selected methods for the 11-level MLI. It is observed that the results are very close to each other in the range of $0.5 \leq M_i \leq 0.8$, while the THD falls below 8%. If Figure 5b and Table 5 are investigated together, it reveals that the GA method is more successful in terms of THD in 11-level MLI. In Figure 5c, the THD performances of the selected methods for the 15-level MLI are given. The performances of all methods are similar and the THD (%) value provides the standard in a wide range, especially in the case of $M_i > 0.45$. However, the performance of the SPBO at low $M_i$ values makes it stand out for 15-level optimization, as indicated in Table 5. Finally, THD performances for 19-level MLI are illustrated in Figure 5d. There is a significant decrease in THD values at low modulation indices due to the decrease in the value of the DC source voltages used and the presence of more switching angles.

At this level, the IEEE 519—2014 standard has started to be met from $M_i = 0.35$. The THD value for BMO, MFO, GWO, and GA methods with a modulation index in the range of 0.65–0.8 decreases below 2%. However, as can be seen in Table 5, the SPBO method provides successful results in 19-level MLI, as well as in the modulation index range of 0.4–0.9 at 15-level.

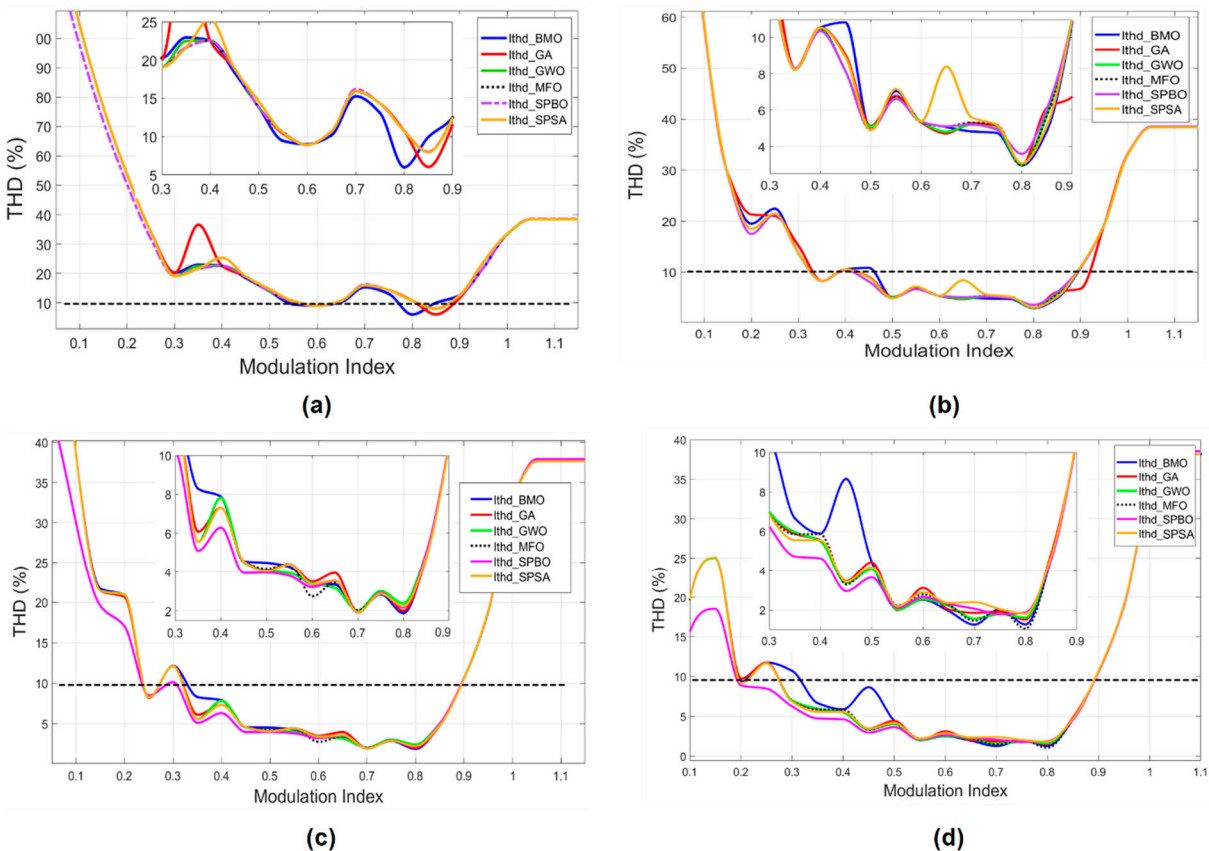

**Figure 5.** THD variations versus modulation indexes for MLI with (**a**) 7-level, (**b**) 11-level, (**c**) 15-level, and (**d**) 19-level applied to suggested metaheuristics.

This study focuses on the elimination of low-order harmonics. Since variations in high-frequency harmonic components for all methods produce similar results, only THD plots of 19-level inverter output current are given in this study. As can be seen from Figure 6, the generation of high-frequency harmonics is quite low compared to the fundamental amplitude.

In Figure 7, the variation of the output voltage as per unit (*pu*) depending on the modulation index is given. The variation of the output voltage for the 7-level MLI is illustrated in Figure 7a. The first thing to notice is that the desired output voltage is reached with a maximum error of 1% at all levels. Since only two harmonic components are optimized in the 7-level MLI, it is clearly seen that the output voltage approaches the desired value very well. It is observed that SPSA and GA methods oscillate compared to other methods at 11-level output voltages in Figure 5b. However, these oscillations are within the tolerance range of 1%. As a result, all methods achieve the desired output voltage. As can be seen from Figure 7c,d, and Table 5, the performances of the methods in 15- and 19-level MLIs become almost the same in terms of output voltage. When Table 5, Figures 5 and 7 are investigated together, similar results emerge in terms of THD and output voltage. Therefore, it is concluded that an inverter level higher than 15 increases the overall cost and the switching losses, although it causes many variations in THD and output voltage.

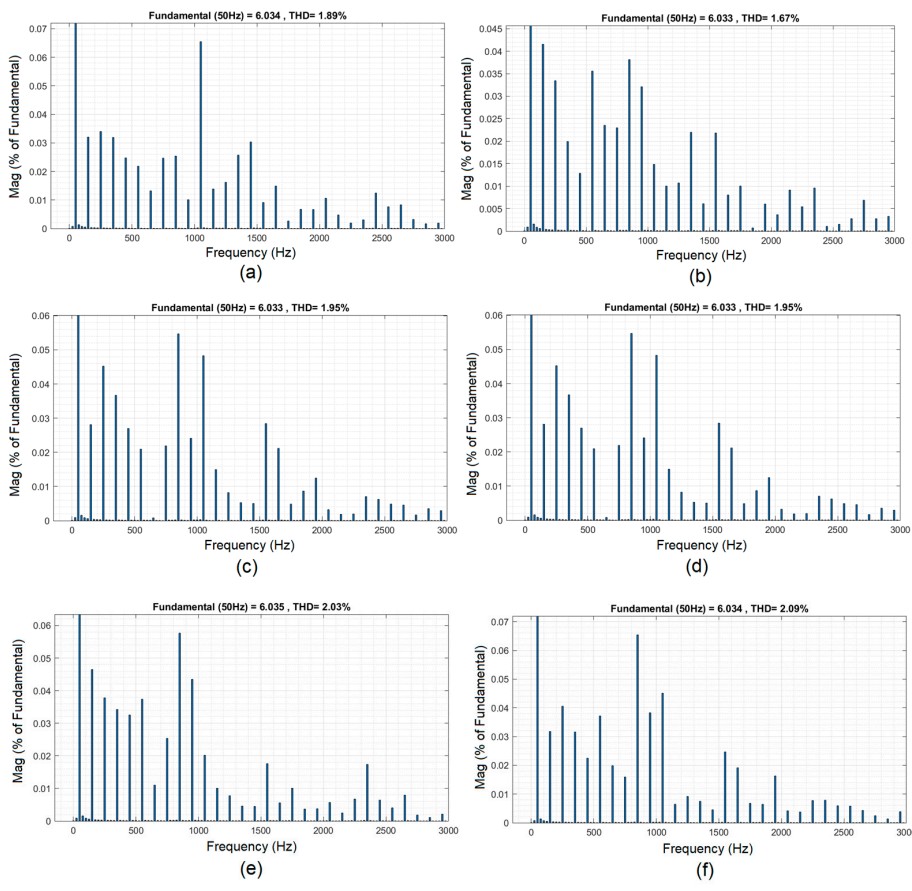

**Figure 6.** FFT analysis of the output current of the 19-level MLI at a modulation index of 0.8 for (**a**) BMO; (**b**) GA; (**c**) GWO; (**d**) MFO; (**e**) SPBO; and (**f**) SPSA.

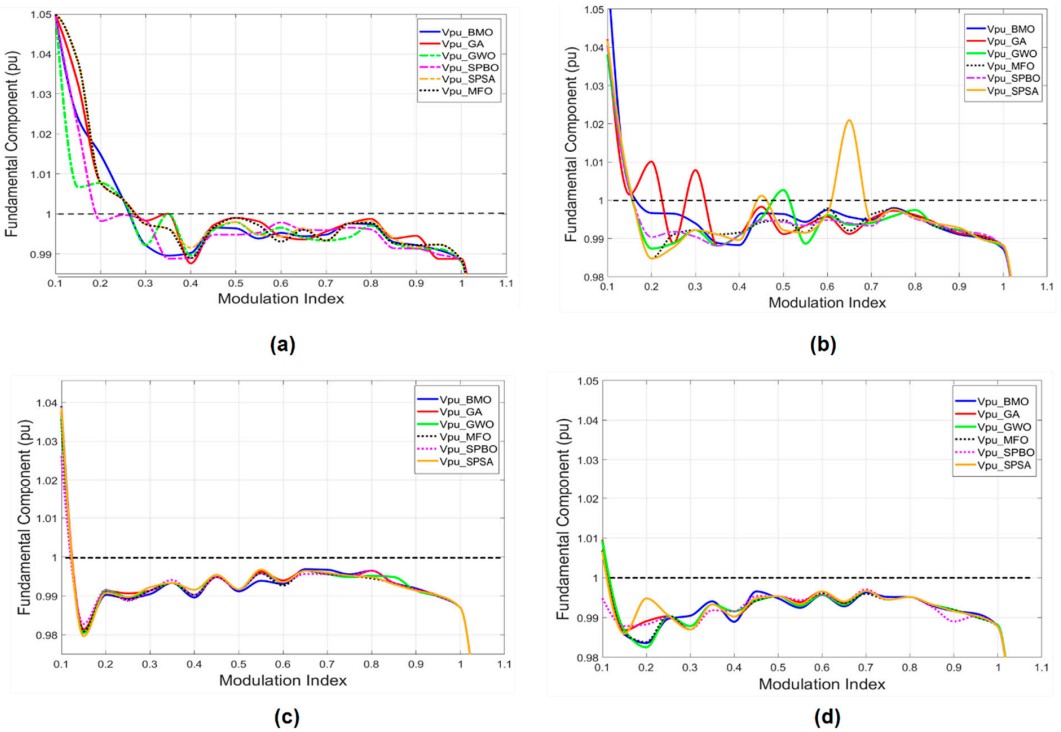

**Figure 7.** The output voltage (per unit) variations versus modulation indexes for MLI with (**a**) 7-level, (**b**) 11-level, (**c**) 15-level, and (**d**) 19-level applied to suggested metaheuristics.



As a result, since the THD value is taken as a reference in the IEEE 519—2014 standard, a method can be recommended for each level among the selected methods according to this reference. No one method makes a significant difference at all levels. In the presented study, it is revealed that the BMO method is superior in 7-level MLI, GA in 11-level, and SPBO method in 15- and 19-level MLIs for the fitness function given by Equation (10).

The graphs of variation of the output current, the output voltage, and harmonic components are plotted for different levels of MLIs considering the proposed methods. In order to obtain the same output voltage at each level, the DC source voltage value is reduced. As can be seen in Figure 8, the output voltage is gained at the desired levels for each modulation index.

Figure 8 shows the current and voltage waveforms for a 7-level MLI, whose switching moments were obtained by the BMO method at 0.2, 0.4, 0.8, and 1.0 values of the modulation index. As can be seen from the figure, as the output voltage approaches the square wave operating mode in the case of $M_i \geq 0.9$, the output current also approximates the square waveform.

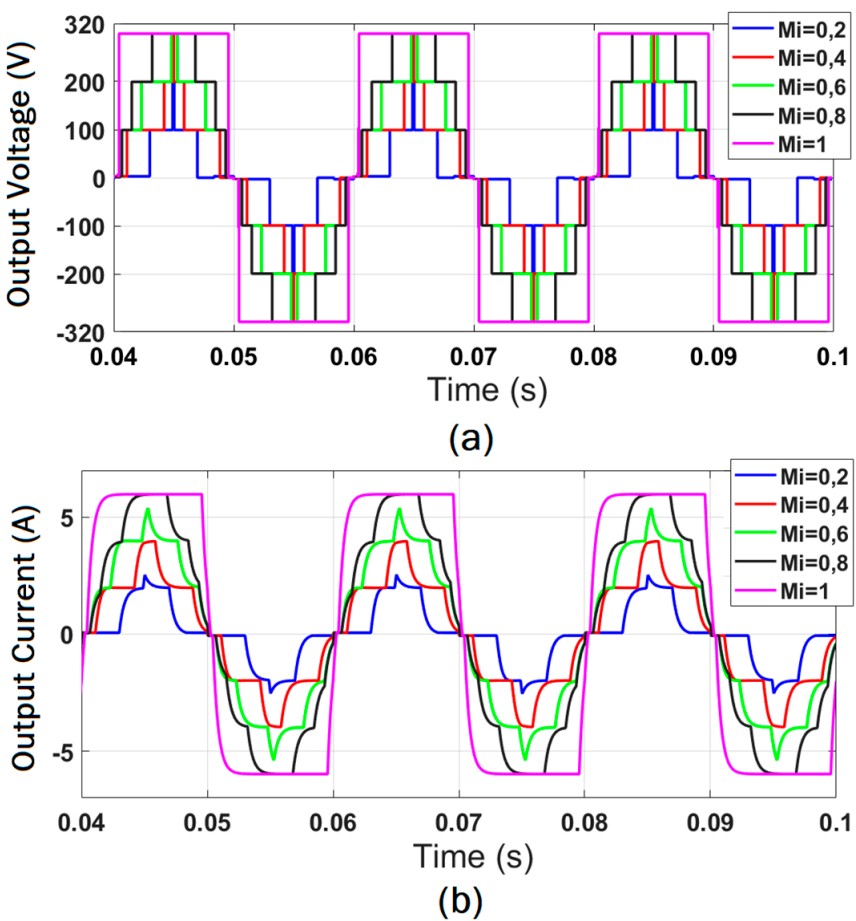

**Figure 8.** The output currents and voltages depend on the modulation index for a 7-level MLI inverter using a BMO.

Figure 9 illustrates the 11-level MLI output voltage and current, where the GA method is applied to obtain the switching moments. It is seen that the output current appears to be much more similar to a sine wave than a 7-level MLI.

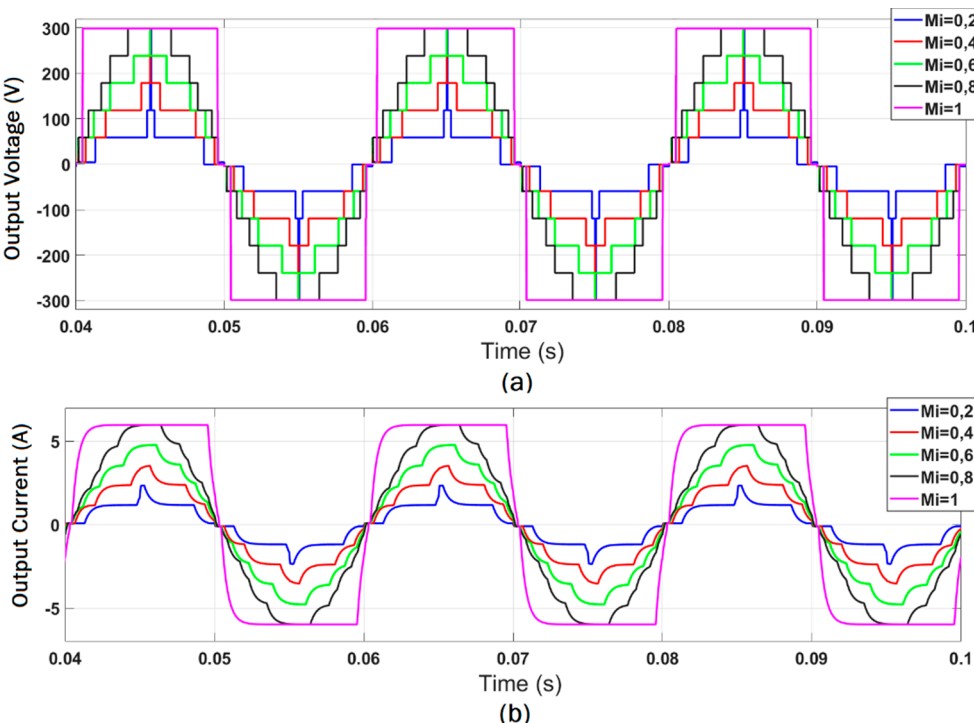

**Figure 9.** The effect of modulation index variations on the output currents and voltages of 11-level MLI employing the GA method.

The output voltages and currents of 15- and 19-level MLIs, where the switching moments are optimized utilizing the SPBO method, are given in Figures 10 and 11. It is clearly seen that the output current approaches the ideal sine waveform as the number of levels increases.

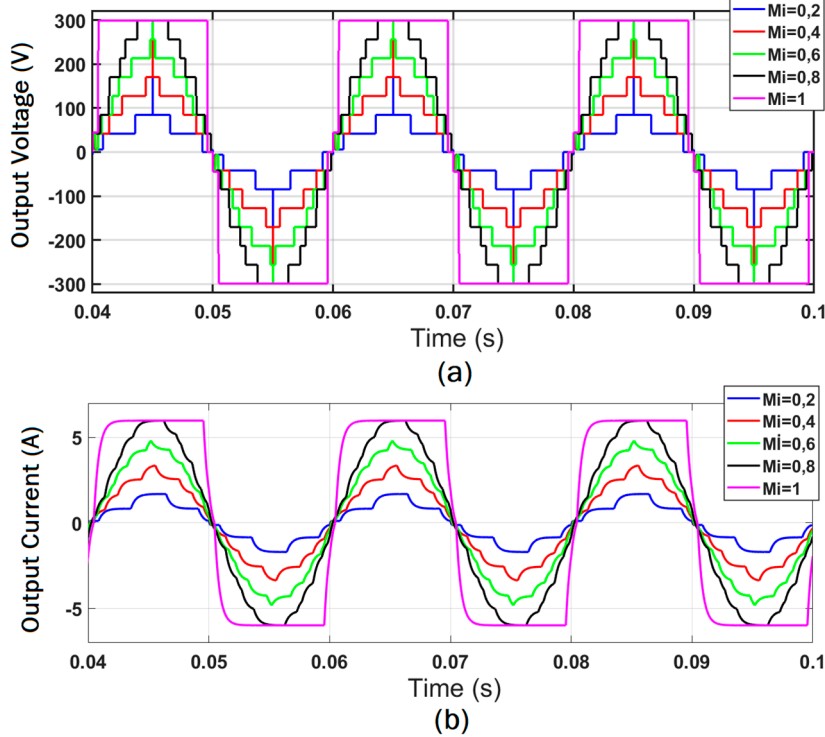

**Figure 10.** The effect of modulation index variations on the output currents and voltages of 15-level MLI employing the SPBO method.

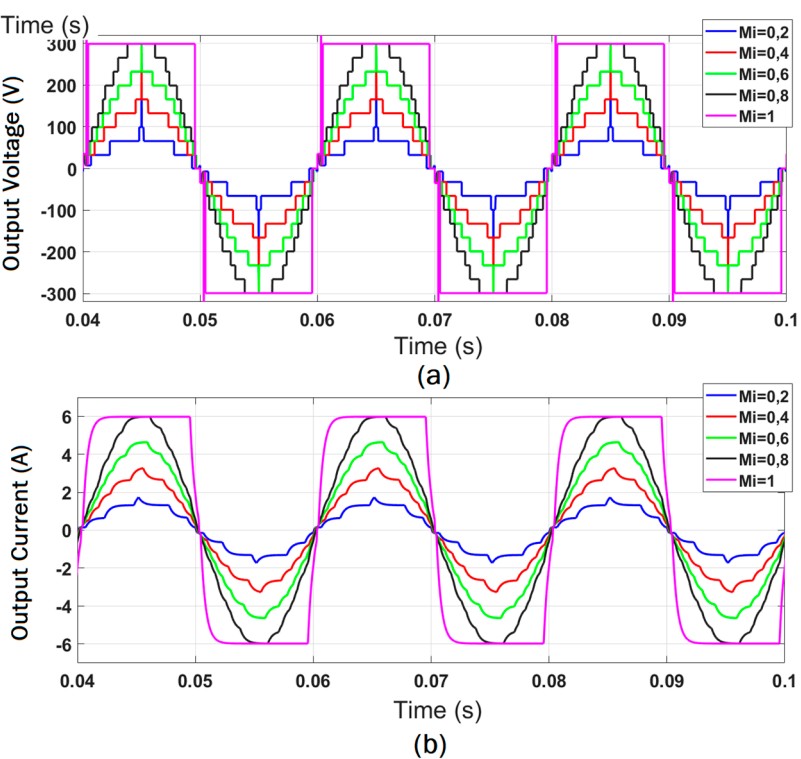

**Figure 11.** The effect of modulation index variations on the output currents and voltages of 19-level MLI employing the SPBO method.

In Figure 12, the variations of the harmonic components depending on the modulation index are demonstrated for 7-, 11-, 15-, and 19-level MLIs. In 7-level MLI, only the 3rd and 5th harmonics appear since the harmonics are eliminated with two of the three DoF formed by a total of three switching signals. This is also valid for other levels.

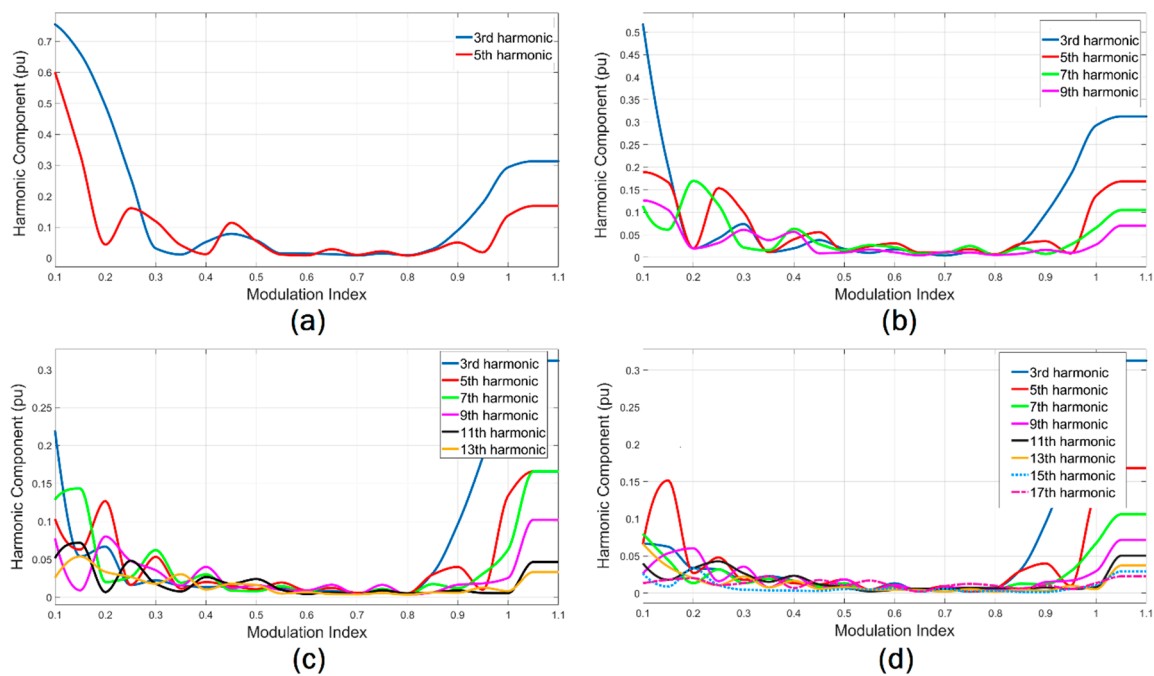

**Figure 12.** Variation of harmonic components depending on modulation index: (**a**) 7-level MLI (BMO); (**b**) 11-level MLI (GA); (**c**) 15-level MLI (SPBO); and (**d**) 19-level MLI (SPBO).

It is clear from Figure 12a that the third and fifth harmonics in the seven-level inverter are removed in the BMO method over a very wide range of modulation indices. Since the number of switching is less in the 7-level MLI compared to the other levels, harmonic components with larger amplitudes occur at low $M_i$ values. In Figure 12b, it is seen that the 3rd, 5th, 7th, and 9th harmonics are eliminated in a wider $M_i$ range compared to the 7-level for the GA method in the 11-level inverter. Since there are five degrees of freedom, four harmonic components can be eliminated. In addition, low-order harmonics were observed at lower values of the modulation index compared to 7-level. However, in case $M_i > 0.9$, the values of the harmonic components increase again, as the operation of the inverter is similar to the square wave mode. Figure 12c describes that the 3rd, 5th, 7th, 9th, 11th, and 13th harmonics are eliminated in a wide range of modulation index values from 0.2 to 0.9 for the SPBO method applied to a 15-level inverter. In the 7-level inverter, the value of the 3rd harmonic, which is 0.7 PU at $M_i = 0.1$, decreases to 0.2 PU. It is clearly seen from Figure 12d that the 3rd, 5th, 7th, 9th, 11th, 13th, 15th, and 17th harmonics are removed over a wide range of the modulation index value from 0.1 to 0.9 for the SPBO method in the 19-level inverter. Another result emerging from the figures is that as the inverter approaches the square wave mode, in the case of $M_i > 0.9$ at all levels, its harmonic components increase and eventually settle at the same values.

## 4. Discussion

In the present study, the effects of 22 metaheuristic algorithms on the solution of the SHE problem in 7-, 11-, 15-, and 19-level MLIs are analyzed. The performances of the recently developed metaheuristics, such as BMO, GBO, GTO, JS, LA, MGA, PO, POA, PPA, SPBO, SPSA, SSO, and WHO, are compared with the widely used GA, SSA, WOA, GWO, PSO, MFO, SCA, DOA, and TLBO.

First of all, the performance of the methods in the 11-level MLI is evaluated in two different ranges of the modulation index (0.1–1.1 and 0.4–0.9), since it would take time and effort to compare all the methods for different levels. As a result of these analyses, the BMO, GA, GWO, SPBO, SSA, and MFO methods among the 22 algorithms mentioned suggest, with their performances, the optimization of the SHE problem. A reanalysis was carried out for the 7-, 11-, 15-, and 13-level MLIs by applying all of the selected methods. Since increasing the number of levels increases the complexity of the fitness function, a single iteration time parameter was also included in the evaluation.

While evaluating the methods in terms of THD, the IEEE 519—2014 standard is taken into consideration. Generally, 7-level MLIs do not meet this standard. However, it is obvious that the BMO method implemented for this problem for the first time in 7-level MLI comes into prominence in terms of THD minimization. On the other hand, it has been revealed that GA, which has been the dominant theorem in optimization for many years, has been successful in 11-level MLI. It was seen that the SPBO method, which was applied to this problem for the first time, was successful in MLI with 15 and 19 levels. Furthermore, 11-level MLI provides this standard in the range of 0.5–0.85 of the modulation index. In MLIs with 15 and 19 levels, this standard is met at the values $0.35 < M_i < 0.85$. According to the IEEE 519—2014 standard, a maximum THD value of 8% is allowed for 1 kW inverters. In terms of THD, since the harmonic elimination in 15-level MLI falls to the desired values, exceeding this level will only increase the cost significantly. By using topologies that employ fewer switching elements, the cost increase can be somewhat reduced.

It has been observed that low-level MLI structures outperform high-level structures in terms of output voltage. This is because the lower weights are assigned to the harmonics in the fitness function. However, as a result of the general evaluation, it is clearly seen that the output voltage error rate generated at all levels stays in the range of 1%. Apart from that, the complexity of the SHE problem increases depending on the number of levels. In the face of this complexity, the variation of all methods in terms of a single iteration time is also different.

Some metaheuristic algorithms apply exploration and exploitation stages to reach the optimal result. These two approaches determine why the algorithm performs poorly or well in an optimization problem. In a metaheuristic algorithm, these two measures must be in balance to reach the result. From this point of view, all methods, except BMO, maintained regular exploration and exploitation throughout their iterations and reached the global optimum. Since the BMO method was trapped at the local optimum, it has not shown the characteristic of decreasing the THD value as the level increases.

As a result, it is concluded that none of the analyzed methods provides optimal results at all levels in terms of both output voltage and THD. Depending on the MLI level, choosing a method appears to be a more appropriate approach.

### 5. Conclusions

In this study, 22 metaheuristic methods for analyzing the SHE problem in 7-, 11-, 15-, and 19-level MLIs were evaluated comparatively. First, the number of metaheuristics was reduced to six with the analysis performed in 11-level MLI with reference to a single iteration time, output voltage quality, and THD minimization parameters. A comprehensive assessment was performed by employing prominent methods, such as SPBO, BMO, GA, GWO, MFO, and SPSA, to 7-, 11-, 15-, and 19-level MLIs according to the IEEE 519—2014 standard.

As a result, the BMO method outperforms in 7-level MLI, GA in 11-level MLI, and the SPBO method in 15- and 19-level MLIs in terms of THD, while in terms of output voltage quality, GA in 7-level MLI, BMO in 11-level MLI, GA and SPSA in 15-level MLI, and SPSA in 19-level MLI come forward. If it is examined in terms of the change in an iteration time depending on the increase in the number of levels, it can be stated that there is an increase in general. Considering the standard deviation value of a single iteration time variation obtained at the different levels, the least change occurs in the GA method, while the SPBO is the method that underwent the most change. If a general evaluation is made, it was observed that any metaheuristic method did not come to the fore at all levels as a result of the analysis. Although THD minimization and output voltage quality increase with the increase in the inverter level, there is not much change after a certain value (15 levels in the present study). Increasing the number of levels further will increase the total cost and switching losses after a certain point.

**Author Contributions:** Conceptualization, S.Ü. and H.Y.; methodology, S.Ü., H.Y. and S.M.; software, S.Ü. and H.Y.; validation, S.Ü. and H.Y.; formal analysis, S.Ü. and H.Y.; investigation, S.Ü., H.Y. and S.M.; resources, S.Ü.; data curation, H.Y.; writing—original draft preparation, S.Ü. and H.Y.; writing—review and editing, S.Ü., H.Y. and S.M.; visualization, S.Ü. and H.Y.; supervision, S.Ü.; project administration, H.Y. All authors have read and agreed to the published version of the manuscript.

**Funding:** This research received no external funding.

**Data Availability Statement:** No new data were created or analyzed in this study. Data sharing is not applicable to this article.

**Conflicts of Interest:** The authors declare no conflict of interest.

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
