# Peer review of "Investigation of Recent Metaheuristics Based Selective Harmonic Elimination Problem for Different Levels of Multilevel Inverters"

_electronics, doi:10.3390/electronics12041058_

Round 1

Reviewer 1 Report

Overall, the work has sufficient merit to be published in Electronics. However, kindly address a few issues: 

(i) The references can be improved by adding 5-8 more works from 2023 and 2022. 

(ii) Please check whether the symbols in equations have been defined or not. 

(iii) Justify the choice of fitness function in Eq (10). 

(iv) Gray Wolf Optimization (GWO) or Grey Wolf Optimization (GWO) ? Please double check by referring to the pioneer work. 

(v) Please add the flowchart of algorithm of this work to improve the readability of this work. 

Overall, interesting work. 

Reviewer 2 Report

1. The term 'proposed' is confusing as the selected MLI already exists in the literature.

2. Why so many algorithms have been compared?

3. The waveform of the MLI can be constructed by careful selection of the steps approximating the required wave, the justification of SHE PWM is required as THD is already low. 

4. The SHE PWM forces the inverter to draw/inject power at higher frequency components, the harmonic spectrum must be presented for each algorithm. There may be the possibility of unwanted harmonics due to switching angles calculated by different algorithms. 

Round 2

Reviewer 2 Report

Thanks for addressing the queries.